# Patent Analysis and Structural Synthesis of Epicyclic Gear Trains Used in Automatic Transmissions

**Huafeng Ding** [1,2,3,*] 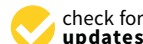 **and Changwang Cai** [1,2]

1    Hebei Provincial Key Laboratory of Parallel Robot and Mechatronic System, Yanshan University, Qinhuangdao 066004, China; cai1990@stumail.ysu.edu.cn

2    School of Mechanical Engineering, Yanshan University, Qinhuangdao 066004, China

3    School of Mechanical Engineering and Electronic Information, China University of Geosciences (Wuhan), Wuhan 430074, China

\*    Correspondence: dhf@ysu.edu.cn

**Abstract:** A patent survey is conducted to investigate the application status of epicyclic gear trains (EGTs) used in automatic transmissions (ATs). In this regard, 673 AT patents are investigated firstly, and 274 different EGTs are obtained after the patent processing. EGTs are presented by corresponding double bicolor graphs (DBGs) and they are sorted into 13 groups in accordance with the degree of freedom (DOF) of DBGs and the degree of hollow vertices. Then, structural characteristics of obtained EGTs, which are used to detect whether the EGTs obtained in the future can be used in ATs, are analyzed and summarized. Finally, a new approach is proposed to obtain new configurations of the EGTs with one main shaft based on the transformation relation between DBG and the basic graph. Several examples are presented to illustrate the validity of the proposed approach.

**Keywords:** epicyclic gear trains; automatic transmission; characteristic analysis; synthesis approach

## 1. Introduction

In the automotive industry, automatic transmissions (ATs) with high efficiency and low consumption are desired to overcome the problems originating from the energy crisis and environmental concerns. Moreover, studies show that vehicles equipped with ATs have remarkable advantages, including simple operation, smooth shift and long service life, which can provide a good driving experience. Since the advent of the first cars equipped with ATs, these vehicles have continuously developed so that they attracted a vast number of consumers, and quickly occupied the dominant place in the field of automotive, especially in U.S., Japan, and European countries [1].

An AT mechanism typically consists of two parts: the hydraulic transmission and mechanical transmission. The hydraulic transmission mainly includes a torque converter, while the mechanical transmission mainly consists of a set of clutches and brakes (hereafter called shifting elements) and a gear train. Operation of the gear train working together with several shifting elements provides various speed ratios, where these speed ratios contribute to the vehicle performance and the fuel economy. Reviewing the literature indicates that gear trains in ATs are mainly divided into two types, namely the ordinary gear train (OGT) and the epicyclic gear train (EGT). It should be indicated that EGTs are more widely used in the AT industry, because they have remarkable superiorities, including the compact structure, larger speed ratios, higher efficiency, and longer working life, over OGTs. Therefore, studying the structural characteristics of EGTs is of great significance to obtain ATs with better performance.

Johnson and Towfigh [2] proposed a synthesis method for EGTs based on the synthesis approach of linkage-type kinematic chains. Moreover, they synthesized gear mechanisms with one degree of

freedom (DOF) and up to eight links. Lévai [3] classified planetary mechanisms in detail and defined the simple planetary gear train (PGT) consisting of two central gears, one or more planet gears and one arm. Buchsbaum and Freudenstein [4], and Freudenstein [5] introduced the graph theory into the synthesis process of geared kinematic chains (GKCs) and obtained epicyclic gear chains with up to five links. Ravisankar and Mruthyunjaya [6] applied the graph and the matrix theories to synthesize GKCs and obtained all one-DOF EGTs with up to four fundamental loops. Furthermore, Tsai [7] computed the characteristic polynomial of EGTs for the isomorphism detection and synthesized non-isomorphic EGTs with up to six links using the genetic graph approach. Moreover, Tsai and Lin [8] synthesized non-fractionated two-DOF EGTs based on the proposed method. Then, Kim and Kwak [9] synthesized EGTs with up to seven links based on a recursive method, which proved the validity of Tsai's achievements. Hsu [10,11] presented a new graph representation, namely, an acyclic graph representation of PGTs with Lam. Then he proposed a synthesis method for GKCs based on the proposed graph representation [12]. Chatterjee and Tsai [13,14] presented the canonical graph representation of GKCs to solve the pseudo-isomorphic problem, and synthesized EGTs with one-, two- and three-DOF up to eight links. Castillo [15] presented a synthesis method regarding the planets and carriers and 1-DOF EGTs up to nine links were synthesized. Ding et al. [16–18] presented unified topological representation models for planar kinematic chains and synthesized EGTs with up to nine links. Moreover, Rajasri [19] proposed an approach in accordance with the hamming number and generated EGTs with up to seven links. Kamesh [20] utilized the vertex incidence polynomial, proposed a synthesis method and listed one-DOF non-isomorphic EGTs up to 6 links. Shanmukhasundaram et al. [21,22] synthesized the one-DOF EGTs with up to seven links using the concept of building of kinematic units and performed symmetry analysis of EGTs.

Many synthesis methods for EGTs have been proposed so far and subsequently different configurations have been obtained, which are helpful for the innovation of ATs. Up to now, a wide variety of ATs have been proposed, analyzed and produced. In 1939, the Hydra-Matic was invented through the combination of a hydraulic coupling with a three-row EGT, and it was installed on GM Oldsmobile cars [23]. The Hydra-Matic could achieve four forward gears and one reverse gear so that it was considered as the symbol of ATs at that era. Since then, the ATs have rapidly developed. In 1950, Ford company developed a three-gear AT equipped with the torque converter, representing that the AT technology enters the maturity stage. In 1965, AP company produced the first AT installed on the front engine and front drive (FF) vehicles, which promoted the miniaturization of ATs [24]. Then, originating from the fuel crisis in 1970s, ATs with more gears and smaller volumes were designed. The four-gear ATs with an over-speed gear were developed in this regard in 1977. In 1989, five-gear ATs were invented, while 6-gear ATs appeared at the end of the 20th century [25]. Nowadays, ATs with 6–8 gears are mainstream products, while the nine-gear ATs have been developed and applied in some products. Moreover, hybrid transmission systems are studied increasingly common, which are expected to solve the problem of fuel consumption [26–28]. Reviewing the literature indicates that ATs are developed towards more gears, lightweight structure, higher efficiency, and mechanical electronicalization. Figure 1 shows some typical AT products to illustrate the trend.

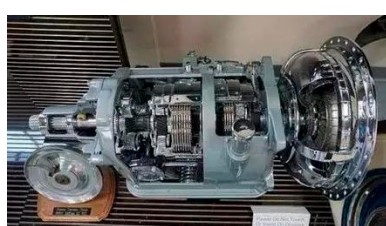
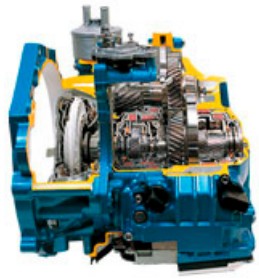
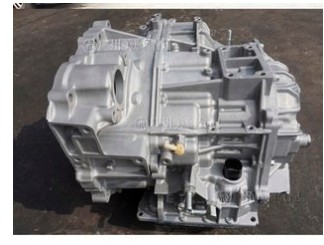

(**a**) GM Hydra-Matic (1939)    (**b**) Aisin TF60SN (2002)    (**c**) Toyota U660E (2006)

**Figure 1.** *Cont.*

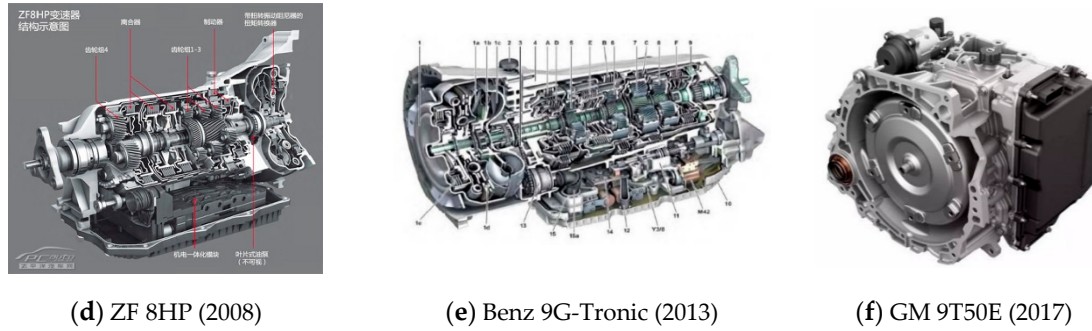

(**d**) ZF 8HP (2008)        (**e**) Benz 9G-Tronic (2013)        (**f**) GM 9T50E (2017)

**Figure 1.** Some typical automatic transmission (AT) products. Note: (**a**) A four-gear AT, (**b**) A six-gear AT, (**c**) A six-gear AT, (**d**) An eight-gear AT, (**e**) A nine-gear AT, (**f**) A nine-gear AT.

Along with the increasing requirements of the driving performance, ATs equipped with EGTs, which have improved configurations and more links, are desired. In order to pick up EGTs more accurately and more suitably, the application status of EGTs used in ATs should be figured out. Therefore, a patent survey is conducted for ATs and the structural characteristics of discoverable EGTs are analyzed and summarized. Studies show that EGTs that violate the structural characteristics should be excluded when synthesizing novel EGTs, and the work efficiency in the design stage of ATs will be improved. Taking the results of the survey and summary as the screening principles, a new method is proposed to investigate the EGTs with one main shaft used in ATs, covering existing designs and new designs. The representation model of EGTs proposed by Ding et al. [16] is applied in the proposed method, which can effectively avoid the pseudo-isomorphism problem.

The overall structure of the present study is as follows. EGTs employed in this article are introduced in Section 2. Then the survey about AT patents and the corresponding results are presented in Section 3. Moreover, the survey is analyzed and the structural characteristics of EGTs are summarized in Section 4. Finally, a new method is proposed in Section 5 to obtain novel configurations of EGTs with one main shaft and several configurations are presented as examples.

## 2. Representation of EGTs

### 2.1. Graph-Based Representation

The topological graph in the graph theory can conveniently represent the topological structures of all kinds of kinematic chains. Based on the unified topological representation models proposed by Ding et al. [16], the topological graph of EGTs can be obtained as follows:

(1) Use solid vertices ("●") and hollow vertices ("○") to denote links and multiple joints, respectively.

(2) Use solid lines ("—") and dash lines ("—") to present revolute pairs and gear pairs, respectively.

It should be indicated that topological graphs of EGTs are double bicolor graphs (DBGs) in the view of the graph theory.

For an EGT, only coaxial links, namely central gears (i.e., sun and ring gears) and carriers, can be used as the input, output, or fixed members to obtain the desired speed ratios [29]. The coaxial links form a multiple joint based on the definition of multiple joint [30], which means that the degree of the hollow vertex is equal to the number of coaxial links. The number of actual revolute pairs at the multiple joint is one fewer than that of coaxial links. There are the following quantitative relationships between EGTs and corresponding DBGs.

$$N_l + N_m = v \tag{1}$$

$$N_r = N_s - N_m \tag{2}$$

$$N_g = N_d, \tag{3}$$

where $N_l$, $N_m$, $N_g$, $N_r$ denote the number of links, multiple joints, gear pairs and revolute pairs in an EGT respectively; $v$, $N_s$, $N_d$ denote the number of vertices, solid lines and dash lines of corresponding DBG respectively.

According to the Chebychev-Grübler-Kutzbach criterion for planar mechanisms [31], the DOF of EGT and corresponding DBG can be obtained.

$$F_E = 3(N_l - 1) - 2N_r - N_g \tag{4}$$

$$F_D = 3(v - 1) - 2N_s - N_d, \tag{5}$$

where $F_E$ denotes the DOF of EGT, $F_D$ denotes the DOF of corresponding DBG.

According to Equations (1)–(5), there is the following relationship between the DOF of EGT and the DOF of DBG.

$$F_E = F_D - N_m. \tag{6}$$

For the EGTs with one main shaft, there is only one multiple joint so that the DOF of corresponding DBGs is one bigger than that of EGTs. For example, the EGT shown in Figure 2a has six links (i.e., labels 2–7) and four of them (i.e., labels 2, 4, 6, 7) are coaxial links forming a multiple joint (i.e., label 1). There are five revolute pairs and four gear pairs in total. According to the Equation (4), the DOF of the EGT can be obtained.

$$F_E = 3 \times (6 - 1) - 2 \times 5 - 4 = 1 \tag{7}$$

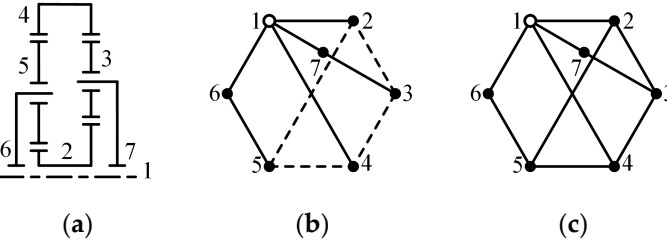

|     |     |     |
| --- | --- | --- |
| (**a**) | (**b**) | (**c**) |

**Figure 2.** (**a**) A six-link epicyclic gear train (EGT) functional diagram, (**b**) the corresponding double bicolor graphs (DBG), (**c**) the basic graph. Note: The numbers in (**a**) and (**b**) represent the corresponding links and multiple joints, while the numbers in (**c**) are in one-to-one correspondence with the numbers in DBG but no practical meaning. The meaning of numbers in different figures is not exactly the same. For example, in this figure, number 1 denotes multiple joint, number 2 denotes sun gear, number 4 denotes ring gear, numbers 3 and 5 denote planet gears, numbers 6 and 7 denote carriers.

According to the Equations (1)–(3), the corresponding DBG has six solid lines, four dash lines and seven vertices including one hollow vertex. Then, the DOF of the DBG can be obtained based on Equation (5).

$$F_D = 3 \times (7 - 1) - 2 \times 6 - 4 = 2 \tag{8}$$

It meets the relationship obtained in Equation (6). The corresponding seven-vertex two-DOF DBG is shown in Figure 2b. By transforming the dash lines into solid lines, DBGs can be transformed into multiple joint topological graphs, which have been synthesized by Ding et al. [32]. In the present study, the multiple joint topological graphs are defined as the basic graphs of DBGs. Figure 2c illustrates the basic graph of the DBG shown in Figure 2b.

*2.2. Link Assortment of DBGs*

The degree of the vertex is defined as the number of lines that are incident with the vertex. Tsai [14] defined the link assortment of mechanisms as a number sequence $[N_2; N_3; N_4; \ldots; N_P]$, where $N_2$, $N_3$,

$N_4, \dots, N_P$ are the number of *i*-degree vertices ($i = 2, 3, 4, \dots, p$). It should be indicated that when $N_i > 9$, capital letters are applied. For example, A and B are associated with 10, and 11, respectively. The concept of the link assortment can be used for the DBGs, and the degree of the vertices of DBGs can be observed intuitively. Usually, the hollow vertices have maximum degrees for DBGs of EGTs with one main shaft.

For example, the link assortment of the DBG shown in Figure 2b is [2; 4; 1], which indicates that there are two vertices of degree two, four vertices of degree three and one vertex of degree four. For the DBGs with *v* vertices and *e* lines, the possible link assortments can be obtained through the following expressions.

$$N_2 + N_3 + \cdots + N_p = v \tag{9}$$

$$2N_2 + 3N_3 + \cdots + pN_p = 2e. \tag{10}$$

## 3. Research of AT Patents

### 3.1. Search Strategy

In order to investigate the existing designs of EGTs used in AT technology, a comprehensive patent survey of ATs is conducted from the graph theory point of view. The purpose of this section is to investigate the structural characteristics of EGTs, which could be taken as the screening principles in the design stage of ATs.

The scoping review is performed for patents based on PRISMA (preferred reporting items for systematic reviews and meta-analyses) guidelines [33]. Considering the convenience of understanding and retrieval, the China patents are chosen as the main searching object. Moreover, more and more foreign companies pay attention to the Chinese market and apply for patent protection in China, which ensures the comprehensiveness of our survey. Therefore, the patent searching is conducted mainly in the SooPAT database and supplemented in the Espacenet database, because the former one includes almost whole patents applied in China while the later one includes many patents applied in other countries in English version, which can guarantee the validity of the survey. The keywords for searching are "automatic transmission", "epicyclic gear train", and "planetary gear train". The range of years for searching in the SooPAT database is from 1992 to 2018, and in the Espacenet it is from 1940 to 2018. It should be indicated that the patents applied in recent years are searched although they are not granted yet.

Patents with the following conditions are ignored.

(1) The content is not about the configuration of AT mechanisms. For example, the ones relating to the control system and appearance design are ignored.

(2) The AT mechanisms are equipped with OGTs.

(3) The EGT used in a patent is not a unified system, but formed by two or more EGTs connecting to each other with shifting elements.

### 3.2. Patent Processing

According to the graph-based representation, all EGTs in the patents are transformed into the corresponding DBGs. Therefore, the structural characteristics of EGTs can be obtained by analyzing the corresponding DBGs.

For an EGT with double planet gears, it does not change the mobility, namely the DOF, of the EGT, the value of gear ratios and the number of coaxial links if removing the second planet gear. Therefore, the double-plane EGTs are simplified by removing one of the planet gears. In this regard, Figure 3 illustrates an example to show the abovementioned simplification.

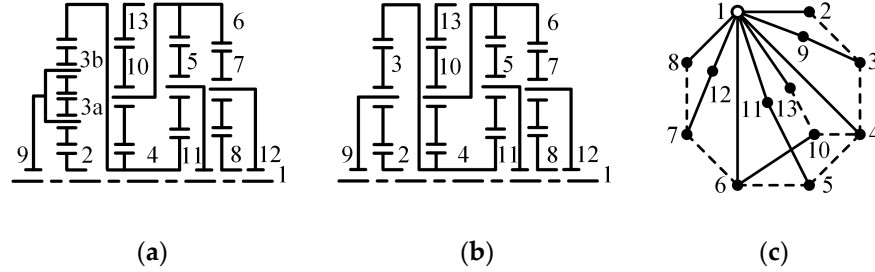

|   |   |   |
|:-:|:-:|:-:|
| (**a**) | (**b**) | (**c**) |

**Figure 3.** (**a**) A double-planet EGT, (**b**) the simplified configuration, (**c**) the corresponding DBG.

It should be indicated that only the configurations of EGTs used in existing designs are considered to be investigated in the present study. The arrangement of components for input, output, and fixed members is not under consideration. The retrieved patents are disposed of as follows.

(1) Obtain EGTs used in the patents by removing housings, torque converters, and shifting elements.

(2) Transform the EGTs into corresponding DBGs.

(3) Categorize the DBGs based on the number of vertices, the mobility and link assortments.

(4) Detect the isomorphism of DBGs in the same group, and only retain one of the isomorphic DBGs.

(5) Transform the retained DBGs into basic graphs.

(6) Detect the isomorphism of basic graphs and only retain one of the isomorphic basic graphs.

### 3.3. Search Results

Searching the SooPAT and the Espacenet databases results in 673 AT patents equipped with EGTs. These results are analyzed based on the search strategy. It should be indicated that 549 patents are obtained from the SooPAT database, while the other 124 patents are supplemented from the Espacenet database. Figures 4 and 5 show the annual number of patents on ATs and the share of each country distribution in patent assignees, respectively. Figure 4 indicates that the number of AT patents in every five years is increasing as time goes on, which means that more and more attention is paid to the innovation of ATs. Figure 5 indicates that about 17.8 percent of the searched patents belong to China, while most of the rest belong to developed countries, such as Germany, U.S., Japan and Korea.

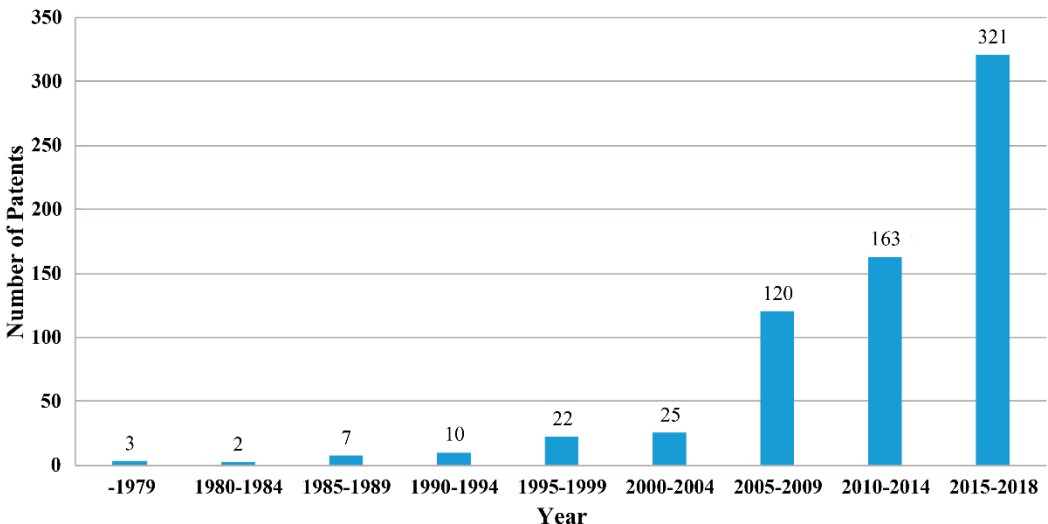

**Figure 4.** Number of annually published patents. Note: The value on the bar graph represents the number of published AT patents in the respective year intervals.

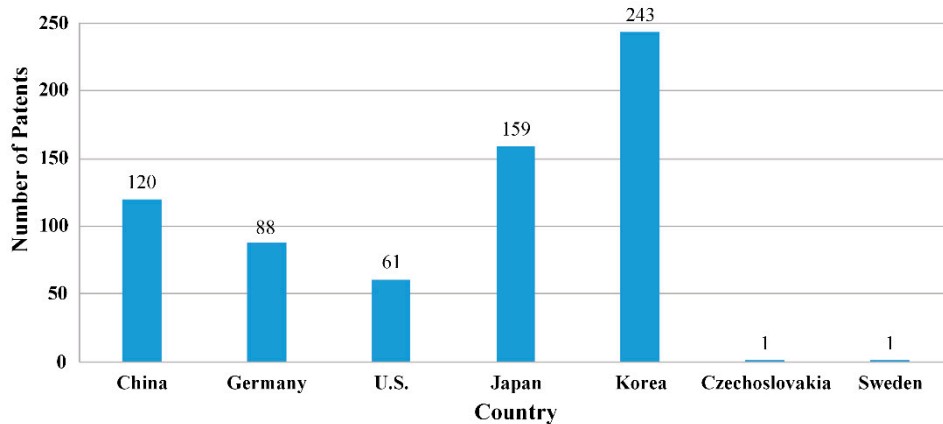

**Figure 5.** Comparison of some country's share in publishing AT patent assignees. Note: the value on the bar graph represents the number of published AT patents in each country.

After patent processing, 274 DBGs and 67 basic graphs are obtained. To categorize and screen DBGs conveniently and intuitively, they are sorted into 13 groups in accordance with the mobility of the corresponding DBGs and degree of hollow vertices. Some of the patents with more than one configuration of EGTs are put into different groups. Table 1 shows the results of the conducted survey.

**Table 1.** Groups of the double bicolor graphs (DBGs) from the survey results.

| Mobility of DBG | Degree of Hollow Vertex | Number of Link Assortments | Number of Basic Graphs | Number of DBG | Number of Patents |
|---|---|---|---|---|---|
| 2 | 4 | 2 | 2 | 5 | 69 |
| | 5 | 5 | 5 | 13 | 21 |
| | 6 | 5 | 7 | 9 | 11 |
| | 7 | 2 | 2 | 2 | 2 |
| | 8 | 1 | 1 | 1 | 1 |
| 3 | 5 | 1 | 1 | 3 | 23 |
| | 6 | 4 | 7 | 31 | 96 |
| | 7 | 6 | 20 | 62 | 109 |
| | 8 | 1 | 1 | 1 | 1 |
| 4 | 7 | 2 | 2 | 10 | 89 |
| | 8 | 4 | 13 | 105 | 258 |
| | 9 | 3 | 3 | 9 | 5 |
| 5 | 9 | 2 | 3 | 23 | 50 |

Because of the space limitation, only part of the search results are presented here. The whole results, including DBGs and corresponding basic graphs, are presented in the Supplementary Materials.

Group 1: two-DOF DBGs with hollow vertex of degree four

For a two-DOF DBG with a hollow vertex of degree four, the corresponding EGT has one DOF and four coaxial links. There are five DBGs in the group, and they are put into two link assortments. The results are shown in Table 2.

**Table 2.** The results of the two-degree of freedom (DOF) DBGs with hollow vertex of degree four.

| Link Assortment | Patent Assignee | Application Number | Functional Diagram | DBG |
|---|---|---|---|---|
| [2; 4; 1] | Howard W. Simpson [34] | US19550552788 |  |  |
| | Aisin [35] | JP19810083087 |  |  |
| | Wuxi Vocational Institute of Commerce [36] | CN201610394630.0 |  |  |
| | Wuxi Vocational Institute of Commerce [37] | CN201610394749.8 |  |  |
| [3; 2; 2] | Pol Ravigneaux [38] | US19360097377 |  |  |

The first and the fifth EGTs shown in Table 2 are known as the Simpson planetary mechanism and the Ravigneaux planetary mechanism, respectively. These mechanisms are widely used in the AT industry.

Group 2: two-DOF DBGs with hollow vertex of degree five

For a two-DOF DBG with a hollow vertex of degree five, the corresponding EGT has one DOF and five coaxial links. There are 13 DBGs in the group, and they are put into five link assortments. Table 3 shows some of these results.

**Table 3.** Some of the two-DOF DBGs with hollow vertex of degree five.

| Link Assortment | Patent Assignee | Application Number | Functional Diagram | DBG |
|---|---|---|---|---|
| [4; 1; 2; 1] | Great Wall Motor [39] | CN201120174766.3 |  |  |
| [4; 2; 0; 2] | Aisin [40] | JP19950090144 |  |  |
| [3; 3; 1; 1] | Jatco [41] | JP20020207330 |  |  |
| [2; 5; 1; 1] | Honda [42] | JP19950041377 |  |  |
| [3; 3; 2; 1] | Toyota [43] | JP19930302363 |  |  |

Group 3: three-DOF DBGs with hollow vertex of degree five

For a 3-DOF DBG with a hollow vertex of degree five, the corresponding EGT has two DOFs and five coaxial links. There are three DBGs in the group, where they belong to a one link assortment. Table 4 shows the results for such DBGs.

**Table 4.** The results of the three-DOF DBGs with hollow vertex of degree five

| Link Assortment | Patent Assignee | Application Number | Functional diagram | DBG |
|---|---|---|---|---|
| [4; 3; 0; 1] | Hyundai [44] | KR19990027509 | | |
| | Jatco [45] | US19910698128 | | |
| | Hyundai [46] | KR19950048216 | | |

Group 4: two-DOF DBGs with hollow vertex of degree six

For a two-DOF DBG with a hollow vertex of degree six, the corresponding EGT has one DOF and six coaxial links. There are nine DBGs in the group, and they are put into five link assortments. Table 5 illustrates some of these results.

**Table 5.** Some of the two-DOF DBGs with hollow vertex of degree six.

| Link Assortment | Patent Assignee | Application Number | Functional Diagram | DBG |
|---|---|---|---|---|
| [5; 1; 1; 1; 1] | Great Wall Motor [47] | CN201210369889.1 | | |
| [2; 6; 1; 0; 1] | Jatco [48] | JP19940133754 | | |
| [2; 7; 0; 1; 1] | Chen Boheng [49] | CN01115089.0 | | |

**Table 5.** *Cont.*

| Link Assortment | Patent Assignee | Application Number | Functional Diagram | DBG |
|---|---|---|---|---|
| [2; 6; 2; 0; 1] | Chongqing Wangjiang Ind Co Ltd. [50] | CN201510878451.X |  |  |
| [3; 4; 3; 0; 1] | Jatco [51] | JP19940103426 |  |  |

Group 5: three-DOF DBGs with hollow vertex of degree six

For a three-DOF DBG with a hollow vertex of degree six, the corresponding EGT has two DOFs and six coaxial links. There are 31 DBGs in the group, and they are put into four link assortments. Some of these results are shown in Table 6.

**Table 6.** Some of the three-DOF DBGs with hollow vertex of degree six.

| Link Assortment | Patent Assignee | Application Number | Functional Diagram | DBG |
|---|---|---|---|---|
| [5; 2; 1; 0; 1] | Hau Antonin, et al. [52] | CS19810006810 |  |  |
| [3; 6; 0; 0; 1] | Nissan Motor [53] | JP19910208859 |  |  |
| [4; 4; 1; 0; 1] | Toyota [54] | JP19780018112 |  |  |
| [5; 2; 2; 0; 1] | Jatco [55] | JP20130064598 |  |  |

Group 6: two-DOF DBGs with hollow vertex of degree seven

For a two-DOF DBG with hollow vertex of degree seven, the corresponding EGT has one DOF and seven coaxial links. There are two DBGs in the group, and they are classified into two link assortments. The results are presented in Table 7.

**Table 7.** Results of the two-DOF DBGs with hollow vertex of degree seven.

| Link Assortment | Patent Assignee | Application Number | Functional Diagram | DBG |
|---|---|---|---|---|
| [3; 7; 1; 0; 1; 1] | ZF [56] | DE19863613331 |  |  |
| [2; 8; 1; 1; 0; 1] | Volvo [57] | EP19840901384 |  |  |

Group 7: three-DOF DBGs with hollow vertex of degree seven

For a three-DOF DBG with a hollow vertex of degree seven, the corresponding EGT has two DOFs and seven coaxial links. There are 62 DBGs in the group, and they are put into six link assortments. Some of these results are illustrated in Table 8.

**Table 8.** Some of the three-DOF DBGs with hollow vertex of degree seven.

| Link Assortment | Patent Assignee | Application Number | Functional Diagram | DBG |
|---|---|---|---|---|
| [5; 3; 2; 0; 0; 1] | Jatco [58] | JP19940075360 |  |  |
| [5; 4; 1; 1; 0; 1] | Hyundai [59] | KR20090059087 |  |  |
| [2; 9; 0; 0; 0; 1] | GM [60] | US20110987799 |  |  |

**Table 8.** *Cont.*

| Link Assortment | Patent Assignee | Application Number | Functional Diagram | DBG |
|---|---|---|---|---|
| [3; 7; 1; 0; 0; 1] | Hyundai [61] | KR20040071080 | | |
| [4; 5; 2; 0; 0; 1] | Toyota [62] | JP19930308666 | | |
| [4; 6; 0; 1; 0; 1] | Aisin and Toyota [63] | JP20140507555 | | |

Group 8: four-DOF DBGs with hollow vertex of degree seven

For a four-DOF DBG with a hollow vertex of degree seven, the corresponding EGT has three DOFs and seven coaxial links. There are 10 DBGs in the group, and they are put into two link assortments. Table 9 shows a part of the results.

**Table 9.** Some of the four-DOF DBGs with hollow vertex of degree seven.

| Link Assortment | Patent Assignee | Application Number | Functional Diagram | DBG |
|---|---|---|---|---|
| [5; 5; 0; 0; 0; 1] | Toyota [64] | EP19890311783 | | |
| [6; 3; 1; 0; 0; 1] | Nissan Motor [65] | JP19830196733 | | |

Group 9: two-DOF DBGs with hollow vertex of degree eight

For a two-DOF DBG with a hollow vertex of degree eight, the corresponding EGT has one DOF and eight coaxial links. Since there is only one DBG in the group, it belongs to one link assortment. The link assortment and the DBG are shown in Table 10.

**Table 10.** The two-DOF DBG with a hollow vertex of degree eight.

| Link Assortment | Patent Assignee | Application Number | Functional Diagram | DBG |
|---|---|---|---|---|
| [2; A; 1; 0; 1; 0; 1] | Volvo [57] | EP19840901384 | | |

**Group 10: three-DOF DBGs with hollow vertex of degree eight**

For a three-DOF DBG with a hollow vertex of degree eight, the corresponding EGT has two DOFs and eight coaxial links. Similar to the previous table, since there is only one DBG in the group, it belongs to one link assortment. The link assortment and the DBG are shown in Table 11.

**Table 11.** The 3-DOF DBG with a hollow vertex of degree eight.

| Link Assortment | Patent Assignee | Application Number | Functional Diagram | DBG |
|---|---|---|---|---|
| [3; 8; 2; 0; 0; 0; 1] | China North Vehicle [66] | CN201510080726.5 | | |

**Group 11: four-DOF DBGs with hollow vertex of degree eight**

For a four-DOF DBG with a hollow vertex of degree eight, the corresponding EGT has three DOFs and eight coaxial links. There are 105 DBGs in the group so that they are divided into four link assortments. Table 12 illustrates some of these results.

**Table 12.** Some of the four-DOF DBGs with hollow vertex of degree eight.

| Link Assortment | Patent Assignee | Application Number | Functional Diagram | DBG |
|---|---|---|---|---|
| [4; 8; 0; 0; 0; 0; 1] | ZF [67] | DE20051002337 | | |
| [5; 6; 1; 0; 0; 0; 1] | Hyundai [61] | KR20040071080 | | |

**Table 12.** *Cont.*

| Link Assortment | Patent Assignee | Application Number | Functional Diagram | DBG |
|---|---|---|---|---|
| [6; 5; 0; 1; 0; 0; 1] | ZF [68] | DE201310205377 | | |
| [6; 4; 2; 0; 0; 0; 1] | Hyundai [69] | KR20050118355 | | |

Group 12: four-DOF DBGs with hollow vertex of degree nine

For a four-DOF DBG with a hollow vertex of degree nine, the corresponding EGT has three DOFs and nine coaxial links. There are nine DBGs in the group, which are divided into three link assortments. Some of the results are shown in Table 13.

**Table 13.** Some of the four-DOF DBGs with hollow vertex of degree nine.

| Link Assortment | Patent Assignee | Application Number | Functional Diagram | DBG |
|---|---|---|---|---|
| [4; 9; 1; 0; 0; 0; 1] | Hyundai [70] | KR20140180671 | | |
| [3; B; 0; 0; 0; 0; 1] | GM [71] | US20050260875 | | |
| [5; 7; 2; 0; 0; 0; 1] | Hyundai [72] | KR20140179663 | | |

Group 13: five-DOF DBGs with hollow vertex of degree nine

For a 5-DOF DBG with a hollow vertex of degree nine, the corresponding EGT has four DOFs and nine coaxial links. There are 23 DBGs in the group, which are divided into two link assortments. Some of the results are shown in Table 14.

**Table 14.** Some of the five-DOF DBGs with hollow vertex of degree nine.

| Link Assortment | Patent Assignee | Application Number | Functional Diagram | DBG |
|---|---|---|---|---|
| [6; 7; 0; 0; 0; 0; 0; 1] | GM [73] | US20080045802 | | |
| [7; 5; 1; 0; 0; 0; 0; 1] | GM [74] | US201113090117 | | |

## 4. Structural Characteristics of EGTs

The existing EGTs used in AT patents obtained from the survey are presented in Section 3, and part of the results are presented by Tables 2–14. In the present study, the number of links of EGTs used for ATs is six to fourteen and the number of coaxial links is four to nine. Moreover, the mobility of EGTs used in ATs is 1–4 and the number of gear rows is 2–6. ATs are developed towards more gears, which requires the number of links of EGTs, especially coaxial links, becoming more and more. In accordance with the mobility of DBGs and the degree of hollow vertices, the results are sorted into 13 groups. Furthermore, the DBGs in the same groups are divided into several subgroups based on the link assortments they belong to. People can quickly determine which group a DBG belongs to and detect whether the DBG is a new configuration at the concept stage of the design process for ATs. Through analyzing the functional diagrams of EGTs and the corresponding DBGs obtained from the patent survey, the structural characteristics of EGTs are summarized as follows.

(1) Table 1 shows that EGTs with the mobility of two and three are widely used in ATs. In order to obtain a certain gear ratio, some shifting elements are engaged to make the mobility of AT mechanism to be equal to the number of input members. If there are *f* shifting elements engaged, an *F*-DOF EGT will be transformed in a (*F*-*f* + 1)-DOF AT mechanism. For example, the structure diagram of an AT mechanism [59] is shown in Figure 6a, which consists of 6 shifting elements and a 11-link two-DOF EGT as shown in Figure 6b. There is one input member in the AT mechanism. Therefore, the number of engaging shifting elements should be (2 + 1 − 1), namely two, under each gear, which is consistent with the arrangement in the patent.

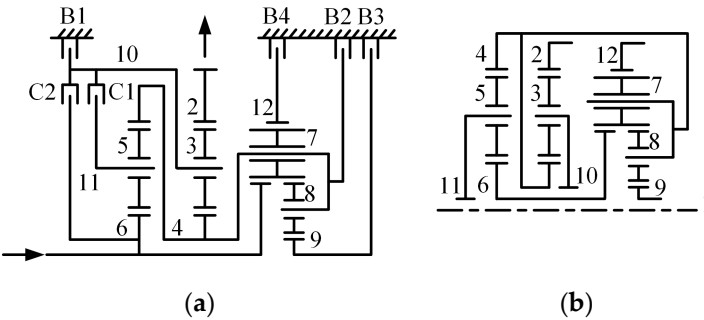

(a)    (b)

**Figure 6.** (**a**) The structure diagram of the AT mechanism, (**b**) the EGT without shifting element engaged.

Generally, there is only one input member in each AT. Moreover, the number of engaging shifting elements have an influence on the complexity and difficulty of the control system. Therefore, the number of engaging shifting elements always be less than four, which means that the EGTs with the

mobility of two or three are more suitable for application in ATs. Furthermore, the input members are always changed to increase the number of gear ratios when one-DOF EGTs are used in ATs.

(2) There are not any redundant links in an EGT. A redundant link is defined by Tsai [14] as the link that is never used as an input, output, or reaction member and does not change the mobility of the mechanism when it is removed. Gear meshing occurs between the planet gear and sun gear (or ring gear) or between planet gears. All of the planet gears in the EGTs from the patent survey are meshed with two or more gears, which means that the number of gear joints for each planet gear is equal to or more than two.

(3) There are not any degenerate structures in an EGT. If there is a group of links rotate at the same velocity in an EGT, there is no relative motion between the links, which is defined as a degenerate structure [75,76]. The DOF of the degenerate structure is zero, namely a rigid chain [77]. The EGT will be transformed into a gear train with fewer links by replaced the degenerate structure with a single link.

(4) The subgraph of the DBG is a tree by removing the dash lines. Figure 7 shows that the dash lines of the DBG of the Simpson planetary mechanism are removed. Moreover, Figure 7b illustrates the subgraph, which is a tree based on the definition in the graph theory.

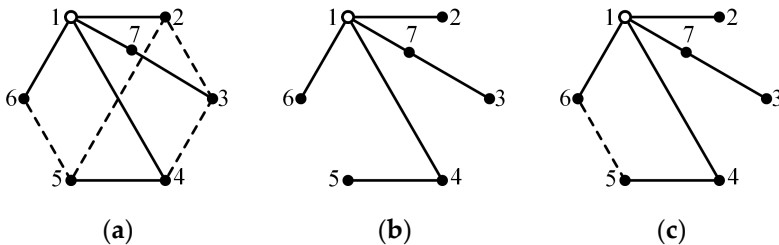

**Figure 7.** (**a**) The Simpson planetary mechanism shown in Table 2, (**b**) the subgraph after removing dash lines, (**c**) the fundamental loop by adding a dash line.

In accordance with the concept of the tree in graph theory, structural characteristics (5) and (6) can be derived.

(5) The number of revolute joints $N_r$ of an $n$-link EGT is equal to $n$-1. According to the graph-based representation in Section 2.1, the solid vertices, the hollow vertices, and the solid lines denote links, multiple joints, and revolute pairs, respectively. Figure 7 indicates that the number of vertices is seven, where one of the vertices is hollow vertex, which means that there are six links and one multiple joints in the EGT. Based on the definition of the multiple joint, the number of revolute pairs should be one smaller than that of solid lines. In other words, the number of revolute pairs is five.

(6) The number of fundamental loops is equal to the number of gear joints $N_g$. The fundamental loops are defined as a loop set that any arbitrary loop of the graph can be expressed as a linear combination of the loops in the loop set. Figure 7 indicates that a fundamental loop is formed by adding a dash line. Therefore, the number of fundamental loops of the Simpson planetary mechanism is 4.

(7) Based on the definition of the DOF for the planar mechanism and structural characteristic (5), the mobility $F_E$ of an $n$-link EGT can be obtained through the expression $F_E = n - 1 - N_g$.

(8) According to the structural characteristic (2), the vertices representing planet gears are incident with a solid line and at least two dash lines so that they cannot connect to the hollow vertex directly.

(9) The vertices representing planet gears would not be adjacent to each other by solid lines. The vertices representing the coaxial links would not be adjacent to each other.

(10) There are loops formed only by gear joints in some EGTs used in ATs. In other words, DBGs including the loops only formed by dash lines can be used in ATs. These EGTs include the fourth EGT shown in Table 2, the second EGT shown in Table 5, and the second EGT shown in Table 7. In the practical design, strict requirements are needed for the size parameter of gears to satisfy the design purpose. Therefore, the structure is usually applied in special situations.

(11) There is one main shaft in each EGT, which means that there is only one hollow vertex in a DBG. Based on the definition of the multiple joints, the lines incident with the hollow vertex should not be presented by dash lines. The hollow vertex has a degree with the maximum number.

(12) Nonplanar DBGs can also be used for ATs, such as the DBG in link assortment [2; 8; 1; 1; 0; 1] shown in Table 7 and the DBG in link assortment [3; B; 0; 0; 0; 0; 0; 1] shown in Table 13.

It should be noted that some of the structural characteristics have been proposed by Tsai [14].

## 5. Method to Obtain New Configuration of EGTs

The performed patent survey results in 274 different DBGs in accordance with the graph-based representation of EGTs. Then, 67 basic graphs are obtained by transforming the dash lines of DBGs into solid lines. Conversely, DBGs, namely the EGTs, can be obtained from basic graphs. Therefore, a method based on basic graphs to synthesize EGTs is proposed here. During the synthesis process, the EGTs violating the above-mentioned structural characteristics should be excluded and the rest can be chosen to use for ATs depending on requirements. According to the source of basic graphs, new configurations of EGTs can be obtained by two means.

### 5.1. Basic Graphs from Existing Products and Patents

Basic graphs can be obtained from existing products and patents through the transformation method. In this way, the probability of obtaining novel configurations similar in structure and performance to existing ones is increased. A lot of preparatory work is needed. The EGTs used in existing AT products and AT patents need to be investigated and be stored in an AT database as completely as possible. These EGTs are not only applied to obtain basic graphs for the synthesis process, but also to detect whether the EGTs obtained by synthesis process have been used before. After the preparatory work, the process of synthesizing new EGTs can be conducted.

(1) Determine the task object. For example, novel six-link one-DOF EGTs with one main shaft are expected to be obtained. There should be one multiple joint, five revolute pairs and four gear pairs according to structural characteristics (5) and (7).

(2) Choose a basic graph of EGTs used in existing AT products or patents based on the task object. The DBGs for the expected EGTs should contain seven vertices, six solid lines and four dash lines, DOF of which is two according to equation (5). The DBGs shown in Table 2 meet the conditions. Take the EGT used in the Ravigneaux planetary mechanism as an example and the corresponding basic graph is obtained as shown in Figure 8.

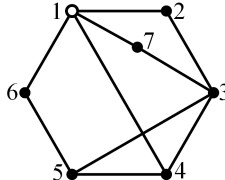

**Figure 8.** The basic graph of EGT used in the Ravigneaux planetary mechanism.

(3) Choose the solid lines that are not incident with hollow vertices. According to the structural characteristic (11), there are 6 solid lines that are incident with the hollow vertex, which means that they can be transformed into dash lines.

(4) Obtain DBGs by transforming solid lines into dash lines. Four of the 6 solid lines should be transformed into dash lines. Then, there should be 15 (i.e., $C_6^4$) DBGs as shown in Figure 9.

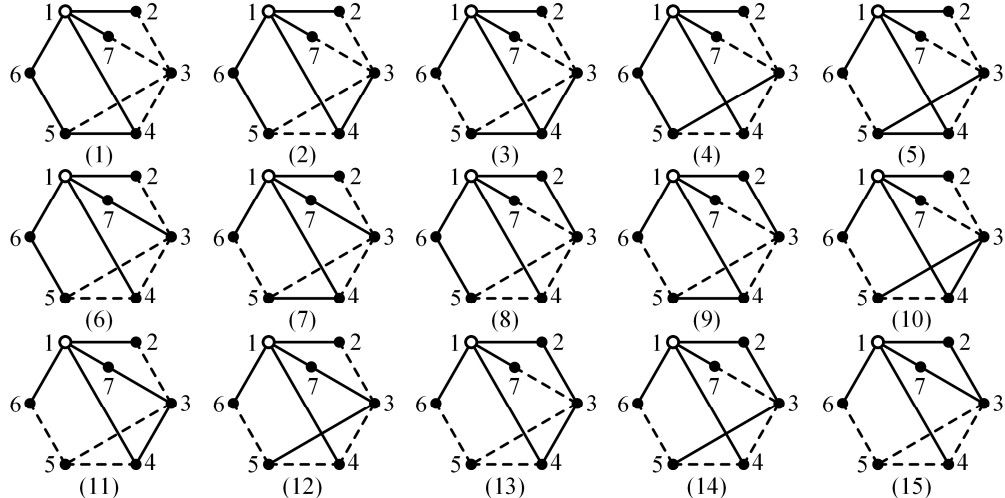

**Figure 9.** The 2-DOF DBGs obtained from the basic graph shown in Figure 8.

(5) Eliminate the DBGs violating the structural characteristics. DBGs (1), (11), (13) and (15) violate the structural characteristic (8), because vertices representing planet gears are incident with dash lines only. On the other hand, DBGs (4), (5), (10), (12) and (14) violate the structural characteristic (9), because the vertices representing planet gears are adjacent with each other by solid lines. Therefore, they should be excluded.

(6) Detect the isomorphism of DBGs. DBGs (6) and (8) are isomorphic with each other, so are DBGs (7) and (9). So, DBGs (6) and (7) are retained only. The retained DBGs are shown in Table 15.

**Table 15.** DBGs obtained from the basic graph and the corresponding functional diagram.

| DBG | Functional Diagram | DBG | Functional Diagram |
| --- | --- | --- | --- |
| 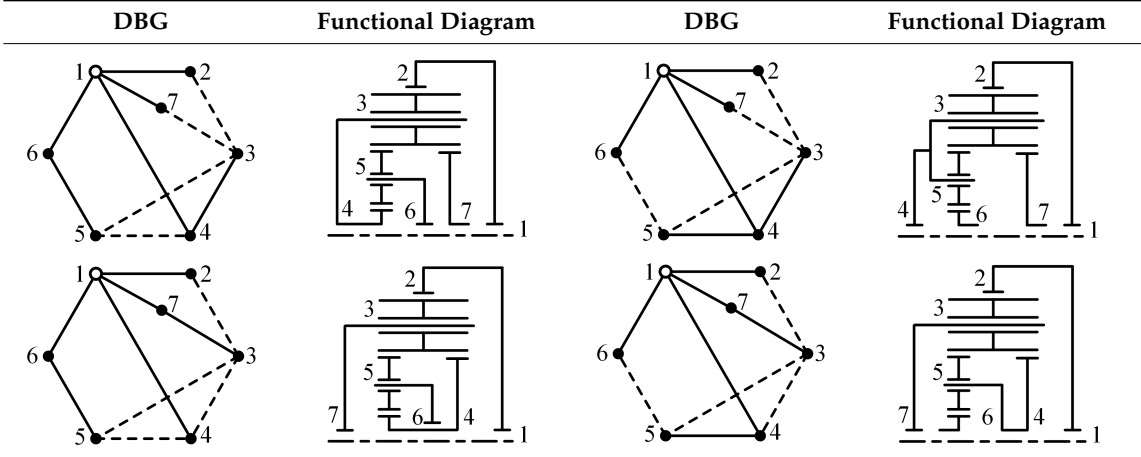 | | | |

(7) Detect whether the retained DBGs have been used in the existing AT products or patents. If not, the DBGs are novel and can be used for innovation of ATs.

The second obtained DBG in Table 15 is the Ravigneaux planetary mechanism, and it is also found in the AT patent registered by Aisin and Toyota companies [78]. In this patent, the AT consists of the Ravigneaux planetary mechanism, a planetary row with double planet gears, 4 clutches, 2 brakes and 1 one-way clutch (OWC). Through the separation and combination of clutches and brakes, it can achieve 8 forward gears and 2 reverse gears. It should be indicated that the patented AT has been applied in LS460 vehicles, named AA80E. The other three DBGs are not found in the AT database established in the patent survey. They are new configurations and new AT products may be obtained by adding several clutches and installing some gear rows. It is found that the third obtained DBG contains a loop

formed by dash lines only. According to structural characteristic (10), it also can be used for ATs in some specific situations.

*5.2. Basic Graphs from Database of One-Multiple-Joint Topological Graphs*

Basic graphs can be obtained from the database of one-multiple-joint topological graphs established by Ding et al. [32], where they synthesized one-multiple-joint topological graphs with up to 16 links. Figure 10 shows some of the one-multiple-joint topological graphs. In this way, the synthesis process is more purposeful and quicker, and configurations totally novel are easily obtained.

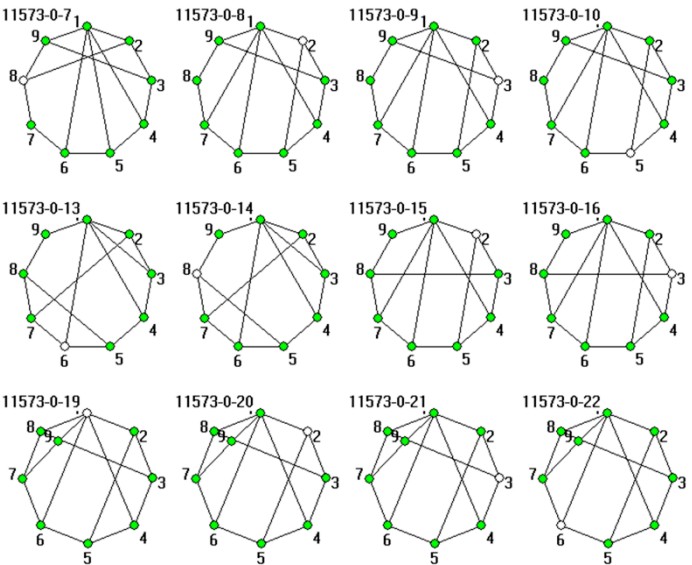

**Figure 10.** Some of the one-multiple-joint topological graphs obtained from the database.

(1) Determine the number of links and mobility of the expected EGTs. For example, EGTs with 8 links and 1 DOF are required for innovation of ATs.

(2) Obtain the number of lines and vertices of expected basic graphs. According to the transformation method, the corresponding DBG has nine vertices and two DOFs. On the other hand, structural characteristics (4)–(7) indicate that there should be 8 solid lines and six dash lines. Then basic graphs applied to be transformed should have eight solid vertices, one hollow vertex and 14 solid lines.

(3) Obtain the mobility of basic graphs. According to the Chebychev–Grübler–Kutzbach criterion for planar mechanisms [31], the mobility of the basic graphs is –5.

(4) Obtain the basic graphs from the database. Based on the database of one-multiple-joint topological graphs, the eight-link (–5)-DOF basic graphs can be obtained. Excluding the graphs violating the structural characteristic (11), the rest of the basic graphs can be used to obtain DBGs. A basic graph in link assortment [1; 7; 0; 1] is chosen as an example, which is presented in Figure 11. Although it is nonplanar, DBGs obtained from it can be applied in ATs in accordance with structural characteristic (12).

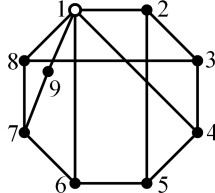

**Figure 11.** An 8-link (-5)-DOF basic graph in the link assortment [1; 7; 0; 1].

(5) Obtain DBGs from the basic graphs. There are 14 lines in the basic graph, where 9 of them can be transformed into dash lines. Therefore, there would be 84 (i.e., $C_9^6$) DBGs. Excluding the DBGs inconsistent with the structural characteristics (2)–(9) and considering the isomorphism problem, there are nine DBGs left. Table 16 shows the remained DBGs. By adding several clutches as appropriate, new configurations can be obtained to use for ATs.

**Table 16.** DBGs obtained from the basic graph and the corresponding functional diagram.

## 6. Conclusions

In order to understand the application status of EGTs used in ATs, a patent survey is conducted and 673 AT patents are analyzed in this regard. Based on the graph-based representation, 274 DBGs are obtained and sorted into 13 groups in accordance with the mobility of DBGs and the degree of hollow vertices. The survey results reveal the application status of EGTs used in ATs, and provide a contrast for the innovation of ATs. Moreover, the structural characteristics of EGTs are analyzed and summarized. The structural characteristics are applied to detect whether the obtained EGTs can be used in ATs and exclude the configurations that violate the requirements of ATs.

Based on survey results and the database of one-multiple-joint topological graphs, by the means of transformation between DBGs and basic graphs, a method is proposed to synthesize EGTs with one main shaft. The representation model, namely DBG, is applied in the proposed method, which can effectively avoid the pseudo-isomorphism problem. Moreover, the proposed method can obtain the configurations similar to the existing ones, as well as the configurations totally novel, which can circumvent the patent protection of existing transmission structure to a certain extent. Furthermore, the method can be implemented by program code and installed on computers to support automatic operations. It is found that the proposed scheme lightens and accelerates the concept stage of the design process.

In the present study, a remarkable number of novel configurations of EGTs are obtained to provide a configurational basis of new ATs with better performance. Obtained novel configurations may provide an opportunity to develop the AT industry. Further work will be considered for the chosen input, output, and fixed members and the arrangement of shifting elements.

**Supplementary Materials:** The following are available online at http://www.mdpi.com/2076-3417/10/1/82/s1: Table S1: The 2-DOF DBGs with hollow vertex of degree four and their basic graphs, Table S2: The 2-DOF DBGs with hollow vertex of degree five and their basic graphs, Table S3: The 3-DOF DBGs with hollow vertex of degree five and their basic graphs, Table S4: The 2-DOF DBGs with hollow vertex of degree six and their basic graphs, Table S5: The 3-DOF DBGs with hollow vertex of degree six and their basic graphs, Table S6: The 2-DOF DBGs with hollow vertex of degree seven and their basic graphs, Table S7: The 3-DOF DBGs with hollow vertex of

degree seven and their basic graphs, Table S8: The 4-DOF DBGs with hollow vertex of degree seven and their basic graphs, Table S9: The 2-DOF DBG with hollow vertex of degree eight and their basic graphs, Table S10: The 3-DOF DBG with hollow vertex of degree eight and their basic graphs, Table S11: The 4-DOF DBGs with hollow vertex of degree eight and their basic graphs, Table S12: The 4-DOF DBGs with hollow vertex of degree nine and their basic graphs, Table S13: The 5-DOF DBGs with hollow vertex of degree nine and their basic graphs.

**Author Contributions:** Conceptualization, H.D. and C.C.; methodology, C.C.; validation, H.D. and C.C.; formal analysis, C.C.; investigation, C.C.; data curation, C.C.; writing—original draft preparation, C.C.; writing—review and editing, C.C.; supervision, H.D.; project administration, H.D.; funding acquisition, H.D. All authors have read and agreed to the published version of the manuscript.

**Funding:** This research was funded by the National Natural Science Foundation of China (Grant No. 51975544 and No. 51675495).

**Acknowledgments:** The authors are grateful to the National Natural Science Foundation of China (NSFC) supporting this research under Contract No. 51975544 and No. 51675495.

**Conflicts of Interest:** The authors declare no conflict of interest.

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
