# Peer review of "Patent Analysis and Structural Synthesis of Epicyclic Gear Trains Used in Automatic Transmissions"

_applsci, doi:10.3390/app10010082_

Round 1

Reviewer 1 Report

It is a very interesting review.
It is carried out very professionally.
I have only been able to observe some small weaknesses

the figure is only named, it would be better to explain or comment at least something else on the figure. Or else remove it.
When you comment on the cheviche-grubler criterion of degrees of freedom, comment that it is 2D. not in 3D.

I consider it a very interesting paper.

Author Response

Dear Reviewer,

Thanks for your comments on our paper. The response to your comments are in the file. Please see the attachment.

Sincerely yours,

Huafeng Ding

Reviewer 2 Report

It is the reviewer's opinion that the work described in the manuscript is well presented and it may be of interest for the research community working in the same field, despite the novelty seems to be limited.

Minor suggestion:

- I would work to improve the quality of the english language (minor mistakes) and the clarity of the presentation

Author Response

(The authors gave the same response as above.)
